# Therapy and Outcome of Prolonged Veno-Venous ECMO Therapy of Critically Ill ARDS Patients

**DOI:** 10.3390/jcm12072499

**Published:** 2023-03-25

**Authors:** Armin N. Flinspach, Florian J. Raimann, Frederike Bauer, Kai Zacharowski, Angelo Ippolito, Hendrik Booke

**Affiliations:** 1Department of Anaesthesiology, Intensive Care Medicine and Pain Therapy, University Hospital Frankfurt, Goethe-University Frankfurt, Theodor-Stern Kai 7, 60590 Frankfurt am Main, Germany; 2Department of Anesthesiology, Intensive Care and Pain Medicine, University Hospital Muenster, University of Muenster, Albert-Schweitzer-Straße 33, 48149 Muenster, Germany

**Keywords:** critical care, acute respiratory distress syndrome, severe acute respiratory syndrome coronavirus 2, extracorporeal membrane oxygenation

## Abstract

Veno-venous Extracorporeal Membrane Oxygenation (VV-ECMO) therapy has become increasingly used and established in many hospitals as a routine treatment. With ECMO-therapy being a resource-demanding procedure, it is of interest whether a more prolonged VV-ECMO treatment would hold sufficient therapeutic success. Our retrospective study included all VV-ECMO runs from 1 January 2020 to 31 June 2022. We divided all runs into four groups (<14 days, 14–27, 28–49, 50+) of different durations and looked for differences overall in hospital survival. Additionally, corresponding treatments and therapeutic modalities, as well as laboratory results, were analyzed. We included 117 patients. Of those, 97 (82.9%) received a VV-ECMO treatment longer than two weeks. We did not find a significant association between ECMO duration (*p* = 0.15) and increased mortality though a significant correlation between the patients’ age and their probability of survival (*p* = 0.02). Notably, we found significantly lower interleukin-6 levels with an increase in therapy duration (*p* < 0.01). Our findings show no association between the duration of ECMO therapy and mortality. Thus, the treatment duration alone may not be used for making assumptions about the prospect of survival. However, attention is also increasingly focused on long-term outcomes, such as post-intensive care syndrome with severe impairments.

## 1. Introduction

Extracorporeal membrane oxygenation (ECMO) was first applied in the 1970s and has become increasingly popular in recent decades as a method of maintaining adequate oxygenation and decarboxylation in critically ill patients [1,2]. Enormous advances in therapeutic regimens supported by advances in circuit design and in cannulation techniques have widened the indications for its use and made the procedure increasingly widespread [3,4]. In addition to the gas exchange of veno-venous ECMO (VV-ECMO), cardiac function replacement was established as a circulatory replacement technique using veno-arterial ECMO (VA-ECMO). In intensive care, the invasiveness of the procedure is still outstanding and is associated with a mortality rate of more than 50% [5]. Other contributing factors are the patients’ mostly critically ill condition, the systemic inflammation process, and the need for (therapeutic) anticoagulation [5,6,7]. This results in organ failure and, in the case of shock or acute renal failure, worsens the prognosis even further (3-month survival rate: 17%) [8,9]. However, this does not diminish the enormous success, resulting in an increase of 200–400% in the number of ECMO treatments in Western countries in the past decade, with VV-ECMO being the predominantly used configuration [5,10].

In the context of the coronavirus (COVID-19) pandemic, health care systems worldwide faced an unprecedented burden of severe COVID-19-associated acute respiratory distress syndrome (CARDS). As a predominantly respiratory disease, CARDS patients received VV-ECMO therapy staring from early on in the pandemic [11]. Similar to prolonged ventilation, VV-ECMO therapy was frequently prolonged for several weeks. An analysis of the ELSO registry data and meta-analysis showed mean treatment durations of 13.9–15.1 days. However, case reports and smaller case series with runs over 100 days have been published [11,12,13,14,15,16,17]. Additionally, a direct comparison of ECMO runs of different viral acute respiratory distress syndrome (ARDS) showed a significantly longer treatment duration in CARDS patients [18]. However, a clear definition of when a VV-ECMO run in general is considered long or prolonged is still lacking. From the data cited, an ECMO run might be described as prolonged after more than two weeks of treatment.

In light of the generally high mortality rate of patients under therapy and the increasing expansion of ECMO runs, it is of interest to determine whether prolonged treatment is worthwhile given the high cost of therapy and the limited resources available. Due to the very limited data on ECMO runs > 14 days, this question is currently unanswered. For this reason, we investigated the outcome and treatment modalities of prolonged ECMO runs performed at our center in recent years.

## 2. Materials and Methods

We conducted a retrospective observational study at a tertiary university hospital, which has adult intensive care units that treat approximately 8200 critically ill patients per year in all specialties, including cardiac surgery. Approximately 3000 patients per year are treated in intensive care units with access to ECMO treatment. The university hospital is connected to an interregional ARDS network and is also recognized as an ELSO center with about 100 ECMO patients annually. The protocol was approved by the institutional ethics committee (#20-643) and registered at clinicaltrials.gov on the 4 April 2022 (NCT05338593). A waiver of written informed consent was approved. The investigators planned and designed this study in accordance with the recommendations of the Declaration of Helsinki and the Strengthening the Reporting of Observational Studies in Epidemiology (STROBE) guidelines, using the suggested checklist [19]. The manuscript adheres to the Consolidated Standards of Reporting Trials (CONSORT) guidelines [20].

### 2.1. Inclusion and Exclusion Criteria

Within the observation period from 1 January 2020 to 31 June 2022, all patients > 18 years treated with VV-ECMO were included in the study. Patients with VA-ECMO were excluded.

### 2.2. Therapy Modalities

We used the cardiohelp^®^ (Getinge AB, Gothenburg, Sweden), rotaflow^®^ (Getinge AB) or rotaflow II^®^ (Getinge AB) systems for VV-ECMO and the Elisa 800^®^ (Löwenstein Medical, Bad Ems, Germany) or Hamilton G5^®^ (Hamilton Medical, Bonaduz, Switzerland) ICU ventilators for mechanical ventilation. All patients received mechanical ventilation, critical care therapy, and ECMO treatment as put forth by Hoyler et al. [21]. For the COVID-19 patients, their therapy adhered to the current recommendations for the treatment of CARDS [22,23,24,25]. Throughout ECMO treatment, oxygenator function was monitored by measuring the daily maximum oxygenation capability (using a gas flow of 12 liters per minute of 100% oxygen via the oxygenator and the resulting p_a_O_2_). To assess mechanical deterioration of the cellular components of the blood, platelet count and lactate dehydrogenase (LDH) were monitored. Occasionally, free hemoglobin and haptoglobin were measured to further quantify hemolysis [26]. Within the context of invasive therapy with VV-ECMO, the clinical protocols aim for a target hemoglobin of 9 mg/dL in order to optimize the availability of systemic oxygen, as well as in the context of bleeding risk under anticoagulation.

Each of the patient’s infection status was assessed by analyzing interleukin-6 (IL-6), leukocyte count, procalcitonin (PCT), and serum lactate dehydrogenase (LDH).

### 2.3. Groups

The patients were categorized according to the duration of ECMO therapy into short ECMO runs of fewer than 14 days, two groups of prolonged runs with a duration from 14 to 27 days, 28 to 49 days, and extremely prolonged runs of more than 50 days.

### 2.4. Data Collection

Clinical data were continuously recorded using a patient data management system (PDMS; Metavision 5.4, iMDsoft, Tel Aviv, Israel). We recorded demographic data, laboratory results, ECMO parameters, ventilation parameters, laboratory findings, and outcomes. For each patient and day, data at 4 a.m., 12 a.m., and 8 p.m. or closest to those times (e.g., for BGA) were extracted by the authors in retrospect.

### 2.5. Statistical Analysis

No statistical power calculation was conducted prior to this study. The sample size was based on the available data. The primary endpoint was defined as in-hospital mortality. Categorical variables are presented as counts and percentages. Nonnormally distributed variables are described as medians (interquartile range, IQR). Demographics and clinical differences between groups were assessed using Fisher’s exact test for categorial variables and the Mann–Whitney U test, as well as the Kruskal–Wallis test, for continuous variables, as appropriate. A van–Elteren test was used to calculate the significant differences in mortality and respiratory minute volume, respiratory rate, tidal volume, and liters per minute (LPM) VV-ECMO blood flow between the groups.

All statistical tests were two-tailed, and results with *p* < 0.05 were considered statistically significant. All calculations/analyses were performed with SPSS^®^ (IBM Corp., Version 26, Chicago, IL, USA).

## 3. Results

Out of the 4510 patients who were treated in our intensive care unit during the observation period, 117 patients received VV-ECMO therapy, of whom all were included in the study (Figure 1). In total, we analyzed 3195 days of treatment with VV-ECMO. Of all patients, 82.9% (*n* = 97) received ECMO treatment for more than two weeks, and 12.8% (*n* = 15) received ECMO treatment for more than 50 days. Two patients received ECMO-treatment for more than 100 days.

Among the participants, we observed a mean BMI of 31.3 kg/m², which corresponds to Class I obesity, according to the International Obesity Task Force (IOTF) cut-off values (Table 1) [27]. The majority of the patients (*n* = 78; 66.7%) were cannulated via femoro-femoral vascular access. The median age was 54 (IQR 8) years, and the patients were predominantly male (*n* = 99; 84.6%).

Regarding the indication for VV-ECMO, the patients had a median p_a_O_2_ of 65.0 (IQR 20.5) mmHg and a median p_a_CO_2_ of 60.8 (IQR 24.2) mmHg while being ventilated with an oxygen fraction of 1.0 (100%) (Table 2). The daily oxygenator capability testing, as described above, had an average p_a_O_2_ of 328 (IQR 115) mmHg. In total, 110 oxygenators were exchanged within the study period. The average runtime of an oxygenator system was 456 (IQR 472) hours (=19 days).

As shown in Table 3, there was high adherence to the target Hb without any difference between the groups (additional data presented in Appendix A). In total, 1441 RBCs (red blood cell concentrates) were administered, which corresponds to one RBC every 2.21 treatment days. There was no difference in the transfusion rate with regard to the duration of treatment. However, we found a statistically significant correlation between the number of RBCs administered (*p* < 0.001, r = 0.042) and the necessity of an oxygenator change. We were unable to find further statistically significant correlations with the need for oxygenator replacement.

Due to the foreseeable long duration of ventilation therapy, a dilatative tracheotomy was performed in the majority of the patients, in 21 (70.0%) patients in the study group of patients up to two weeks of therapy, and in 34 (87.2%) patients, among those treated between two and four weeks. All patients who were treated for longer than four weeks were tracheotomized during the course of treatment. Tracheotomies were conducted under a dilatative technique on the bed side whenever possible (*n* = 114; 97.4%). In our cohort, 51.3% (*n* = 60) of the patients needed renal replacement therapy, and 518 prone positioning maneuvers were performed. The majority of the patients required dual or triple use of sedatives with concomitant opioid-based analgesia.

A detailed presentation of the concomitant clinical characteristics and preexisting medical conditions, as well as complications during the clinical course, can be found in Table 1.

The infection parameter results are displayed in Table 2 and Figure 2A–D, with the corresponding significant differences. Using these laboratory parameters, the sepsis related organ failure assessment (SOFA) score and the clinical assessment of each patient’s condition, 54 patients were diagnosed with sepsis during the course of therapy.

For the ECMO settings (blood flow [L/min] and gas flow [L/min] via the oxygenator), as well as respiratory parameters (tidal volume, respiratory rate, and corresponding respiratory minute volume), no clinically relevant differences were found between the groups (Figure 3, additional data presented in Appendix A).

Statistical analysis showed no significant (*p* = 0.15) correlation between ECMO duration and mortality. However, the two patients with more than 100 days of ECMO treatment died of secondary septic shock. Across all groups, a younger age was associated with a significant (*p* = 0.02) increase in survival (r = 0.21), whereas male sex had no significant impact on mortality (*p* = 0.28). Neither pre-ECMO oxygenation (*p* = 0.75) nor decarboxylation-capabilities (*p* = 0.15) had an effect on survival. Renal replacement therapy and prone positioning also had no effect on survival (*p* = 0.88 and *p* = 0.28).

## 4. Discussion

We conducted a retrospective study of 117 VV-ECMO runs between January 2020 and June 2022 at our tertiary university hospital to assess whether VV-ECMO treatment duration has an impact on mortality. Based on their duration, all runs were categorized into four different groups (<14 days, 14 to 27 days, 28 to 50 days, and >50 days). No significant differences in mortality were found between those groups.

Overall, the treatment of patients with severe ARDS remains very challenging. Although VV-ECMO had already been an established form of therapy for several years, the COVID-19 pandemic resulted in an even further spread of ECMO given the increase in the number of hypoxic patients. However, the in-hospital mortality rate of severe ARDS, which is approximately 50%, remains very high despite the use of VV-ECMO [16,29]. Attempts to explain this observation include the expanded use of VV-ECMO outside of highly specialized centres, with its associated increase in mortality and the question of reasonable implementation [5]. Therefore, the ELSO, for example, has come up with a list of indications and contraindications for the implementation of ECMO therapy [30].

Once, after ECMO treatment is initiated, there are few scores for predicting survival chances from which the RESP (Respiratory ECMO Survival Prediction) score seems to be most promising [31,32]. Given the high technical and personnel requirements, an option for evaluating whether the potential for recovery is decreased, or if it is completely exhausted, is still pending. Particularly in the case of a static treatment course, family members also wish to be informed about the likelihood of the continuation of therapy. However, one aspect that these scores do not consider is the number of days a patient is already on ECMO. Another frequently asked question, either by personnel or relatives, is whether there is a potential for prolonged runtimes to have prospective regeneration/weaning, especially if bridge-to-transplant is not a realistic option. Due to the lack of a clear definition of when an ECMO run is considered prolonged, a scientific analysis of this issue is difficult. Before the COVID-19 pandemic, according to some specialized centers, prolonged therapy is reached after exceeding two weeks. However, due to the experiences gained during the pandemic, the tendency seems to go towards four weeks of ECMO [13,33]. We hence divided our groups accordingly, and our findings did not show any significant differences in mortality or VV-ECMO run duration. Based on our data, ECMO run duration alone is no justifiable factor for limiting treatment.

However, in addition to in-hospital mortality, the long-term outcome of ECMO-treated patients is rightfully becoming an object of scientific interest. The pronounced invasiveness and duration of the treatment are accompanied by a number of serious long-term impairments [34]. Concerning the length of VV-ECMO treatment, it should be considered that the increased treatment length also leads to a higher incidence of secondary disorders such as the postintensive care syndrome, which refers to physical, cognitive, and mental impairments. Few studies have been able to consider the consequences of the particularly invasive ECMO therapy from this perspective. Recently, a small study of 24 young patients reported that 62% of these patients have ongoing fatigue and 47.7% have ongoing pain 8 months after only nine days of ECMO treatment [35].

The rest of our observations support existing data showing that age plays a significant role in survival chances, as Tran and colleagues were able to demonstrate in a study, including 17,449 VV-ECMO runs [29]. While we did not observe their findings of a male gender-associated increase in mortality, this absence of significance (*p* = 0.28) can also be due to our predominantly male cohort.

In our center, a targeted hemoglobin concentration (Hb) for VV-ECMO of approximately 9 g/dL was mostly maintained throughout therapy (mean observed Hb 8.9 g/dL) with 1441 RBC transfusions. Our transfusion practice, thus, appears comparable to the literature, even though lower transfusion thresholds are increasingly being investigated [36]. Our observed correlation (*p* < 0.001, r = 0.042) of RBC transfusion and the need for oxygenator change might be explained by the simultaneous administration of coagulation substitution due to bleeding events, which unfortunately was not recorded separately. However, in the end, the reason for this statistical correlation cannot be answered with our observations, and further targeted studies are needed for explanation. Until then, only hypotheses can be generated.

Based on the sample size of 117 patients and the subsequent split into runtime groups, it was not possible to achieve overall equality in terms of preconditions or demographics. However, only a significant difference between the groups was found for diabetes mellitus, which occurred most frequently in the group with the longest duration (>50 days). Furthermore, cardiovascular diseases also appeared most frequently in the longest ECMO runtime group. Unfortunately, a separate multifactorial regression analysis regarding potential statistical correlation could not be realized.

Our finding of a significantly lower IL-6 and leukocyte count on average with ongoing ECMO treatment may be explained by the frequent occurrence of septic events with their known high mortality. Accordingly, a considerable survival bias may be assumed based on this fact, albeit not leading to a predictive survival benefit.

The possible aetiologies of acute respiratory distress syndrome, as defined by the BERLIN definition, are versatile. The common final pathway of ARDS, leading to VV-ECMO therapy, encompasses a wide range of conditions, from a primary lung injury caused by microbial, viral, chemical, or physical agents, to nonorgan causes, such as severe inflammatory conditions following polytrauma or in the setting of sepsis. As patients often present with advanced lung function impairments, a clear delineation of the aetiology is frequently difficult, since germ detection is not always successful and nosocomial colonization must also be considered a cofounder after a short time. Accordingly, in the present study population, we were not able to define a clear aetiology of ARDS for the majority of the patients; primary organ damage due to in the sense of bacterial causes (e.g., *Streptococcus pneumoniae*, *Haemophilus influenzae*, *Mycoplasma pneumoniae*, etc.) or viral pneumonia (e.g., herpes simplex virus, cytomegalovirus, SARS-CoV-2, etc.) was predominant.

Our study has limitations that need to be taken into consideration when reviewing our results. First, as a mono-centre study from a designated center for extracorporeal therapy the transferability to other hospitals is limited. Second, our study population subsampled VV-ECMO runs based on different aetiologies, which may limit transferability to a specific disease entity. Third, as a retrospective study, no uniform treatment protocols were present, and therapy was conducted according to the attending staff. Additionally, this means that our observed statistical correlations are hypothesis generating and need validation from prospective and targeted studies. Furthermore, it might be considered that the data analysis included an immortal time bias as the patients in the prolonged group(s) only reached the end of the study if they survived until then [37]. Although it is important, that our results do not state that longer ECMO-runs are noninferior to shorter ECMO-runs concerning mortality, the patients who experience prolonged ECMO-runs do have a realistic chance of survival. This is represented by our results, even though an immortal time bias is present.

## 5. Conclusions

Prolonged VV-ECMO runs continue to increase due to the widespread availability of the procedure itself and its concomitant expanded use. Scores for survival prediction can help practitioners. The duration of VV-ECMO treatment does not allow us to assume survival chances from our data. While future analyses may provide more clarity based on growing registry data, for example, from ELSO, our data suggest that prolonged ECMO runs may also appear promising.

## Figures and Tables

**Figure 1 jcm-12-02499-f001:**
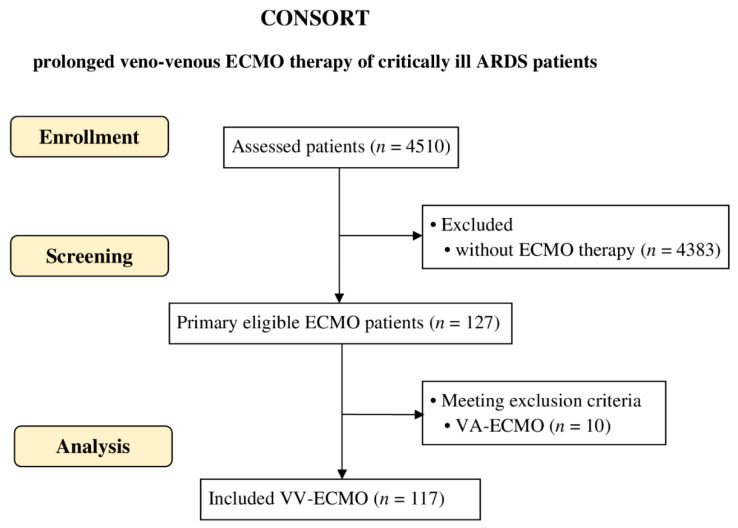
Patient inclusion. Patient inclusion of the study presented as diagram, according to Consolidated Standards of Reporting Trials (CONSORT).

**Figure 2 jcm-12-02499-f002:**
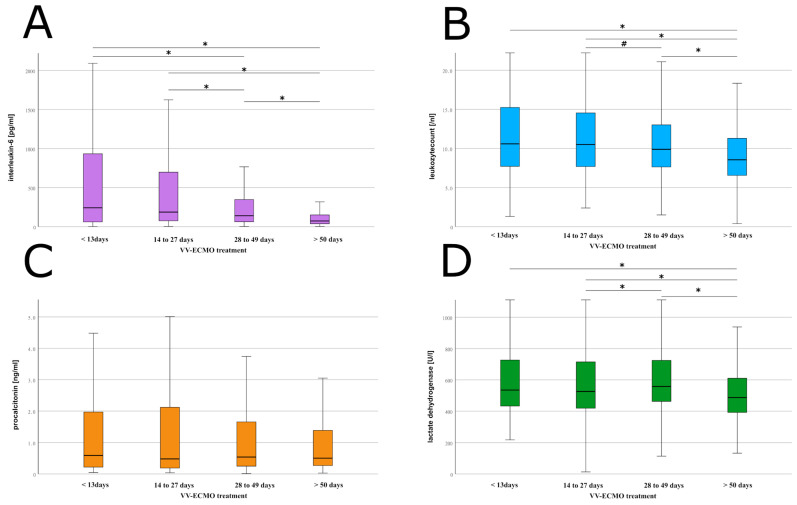
Laboratory infectious parameters. Box and whisker plots of the daily results of the laboratory infectious parameters, subdivided into the included patients with VV-ECMO therapy according to the duration of therapy. (**A**) Illustration of the interleukin-6 values determined twice a day with representation of the significant decrease in values in association with a prolonged VV-ECMO treatment. (**B**) Visualization of the leukocyte count with corresponding significance levels. (**C**) Plots of the daily determined procalcitonin levels, without any significant difference. (**D**) Illustration of the lactate dehydrogenase values and the consequent significant levels. Abbreviations: l, litre; mg milligram; ml, millilitres; ng, nanogram; pg, picogram; U, Units; VV-ECMO, veno-venous Extracorporeal Membrane Oxygenation. * *p* < 0.01, ^#^
*p* < 0.05.

**Figure 3 jcm-12-02499-f003:**
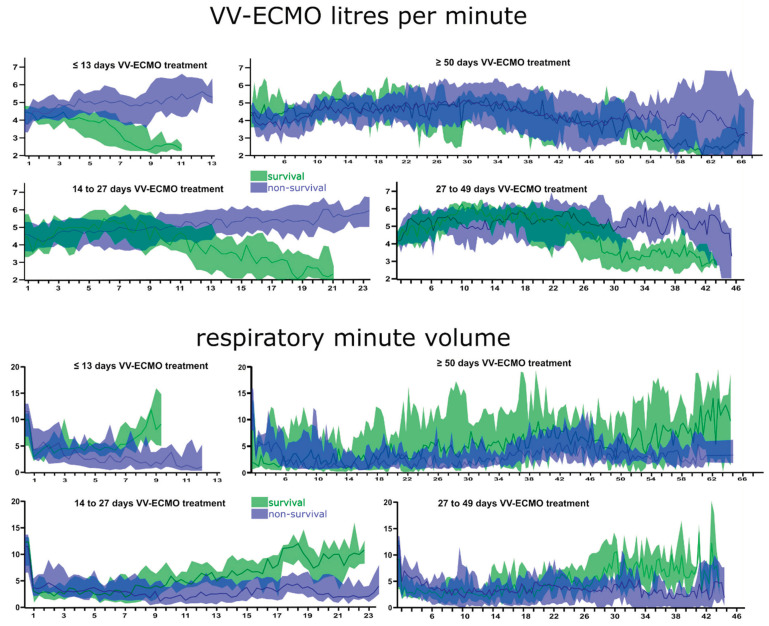
ECMO flow and respiratory minute volume over time separated by survival and grouped by treatment duration. Graphical visualization of the blood flow rates during VV-ECMO therapy and the respiratory minute volume over the course of the treatment days as median, including the corresponding interquantile range as coloured area.

**Table 1 jcm-12-02499-t001:** Clinical characteristics.

Runtime	<14 Days	14–27 Days	28–49 Days	≥50 Days	All
*n* =	30 [25.6%]	39 [33.3%]	33 [28.2%]	15 [12.8%]	117 [100%]
sex (male, %)	27 [90.0%]	32 [82.1%]	28 [84.8%]	12 [80%]	99 [84.6%]
age (years)	52 (IQR: 16)	53 (IQR: 13)	56 (IQR: 20)	56 (IQR: 8)	54 (IQR: 8)
diabetes mellitus	3 [10.0%]	11 [28.2%]	7 [21.2%]	7 [46.7%]	28 [23.9%]
cardiovascular disease	3 [10.0%]	2 [5.1%]	2 [6.1%]	2 [13.3%]	9 [7.7%]
chronic renal failure	3 [10.0%]	1 [3.0%]	1 [3.0%]	0 [0.0%]	5 [4.3%]
arterial hypertension	14 [46.7%]	13 [33.3%]	10 [30.3%]	6 [40.0%]	43 836.8%]
smoking	3 [10.0%]	3 [7.7%]	6 [18.2%]	1 [6.7%]	13 [11.1%]
weight	93.0 (IQR: 10.0)	96.0 (IQR: 15.0)	95.0 (IQR: 18.5)	90.0 (IQR: 20.0)	95.0 (IQR: 14.0)
BMI (kg/m²)	30.5 (IQR: 7.1)	31.0 (IQR: 7.0)	30.5 (IQR: 6.5)	28.0 (IQR: 5.0)	30.5 (IQR: 7.1)
Vjrsvc-Vfrivc *	7 [23.3%]	21 [53.8%]	8 [24.2%]	3 [20.0%]	39 [33.3%]
Vfrivc-Vflivc *	23 [76.7%]	18 [46.2%]	25 [75.8%]	12 [80.0%]	78 [66.7%]

Clinical characteristics and allocation according to therapy duration of the included patients receiving VV-ECMO therapy. Data are presented as median (±interquartile range (IQR)) or as patient number [percentage] where applicable. Abbreviations: BMI, Body mass index; kg, kilogram; m, meters; Vjrsvc–Vfrivc *, draining canulation via right femoral vein with cannula tip in the inferior vena cava and return via cannulation of the right jugular vein with cannula tip ending in the superior vena cava; Vfrivc–Vflivc *, draining canulation via left femoral vein with cannula tip in the inferior vena cava and return via cannulation of the right femoral vein with cannula tip ending in the superior vena cava; VV-ECMO, veno-venous Extracorporeal Membrane Oxygenation. * Cannulation Code, according to the recommendations of the Extracorporeal Life Support Organization (ELSO) Maastricht Treaty for ECLS Nomenclature [28].

**Table 2 jcm-12-02499-t002:** Outcome characteristics.

Runtime	<14 Days	14–27 Days	28–49 Days	≥50 Days	All
*n* =	30 [25.6%]	39 [33.3%]	33 [28.2%]	15 [12.8%]	117 [100%]
Mortality	21 [70.0%]	28 [71.8%]	21 [63.6%]	7 [46.7%]	77 [65.8%]
Ventilation before ECMO	4 (IQR: 7)	1 (IQR: 7)	3 (IQR: 8)	2 (IQR: 7)	3 (IQR: 7)
Any complication	21	27	28	9	85
-Sepsis	12	17	17	8	54
-Delirium	6	10	12	8	36
-Ileus	6	11	12	6	35
-Pulmonary superinfection	7	8	13	4	32
-Intracraniel bleeding	4	1	4	1	10
-Major bleeding	1	9	10	3	23
-Renal failure	11	22	7	8	48

Outcome characteristics of the included patients receiving VV-ECMO therapy. Data are presented as median [±interquartile range [IQR]] or as patient number [percentage] where applicable. Abbreviation: ECMO, Extracorporeal Membrane Oxygenation.

**Table 3 jcm-12-02499-t003:** Therapy characteristics.

Runtime	<14 Days	14–27 Days	28–49 Days	≥50 Days	All
*n* =	30 [25.6%]	39 [33.3%]	33 [28.2%]	15 [12.8%]	117 [100%]
LPM	4.4 (IQR: 1.7)	4.9 (IQR: 1.8)	5.0 (IQR: 2.0)	4.1 (IQR: 2.1)	4.6 (IQR: 1.9)
gas flow *	6.1 (IQR: 4.0)	7.0 (IQR: 3.0)	7.5 (IQR: 2.6)	6.0 (IQR: 3.0)	7.0 (IQR: 3.0)
p_a_O_2_ before ECMO run ^#^	66.0 (IQR: 18.1)	63.9 (IQR: 22.1)	69.8 (IQR: 19.3)	61.0 (IQR: 20.6)	65.0 (IQR: 20.5)
p_a_O_2_ within ECMO run ^#^	71.0 (IQR: 18.1)	69.8 (IQR: 14.9)	67.2 (IQR: 17.0)	69.0 (IQR: 19.0)	69.0 (IQR: 18.0)
p_a_CO_2_ before ECMO run ^#^	62.8 (IQR: 18.7)	60.3 (IQR: 23.3)	56.0 (IQR: 30.7)	67.0 (IQR: 23.0)	60.0 (IQR: 26.9)
p_a_CO_2_ within ECMO run ^#^	48.4 (IQR: 8.3)	48.2 (IQR: 8.3)	50.0 (IQR: 8.9)	52.1 (IQR: 10.0)	49.9 (IQR: 9.8)
Hb [mg/dL]	9.0 (IQR: 1.1)	8.9 (IQR: 1.3)	8.9 (IQR: 1.0)	8.9 (IQR: 0.9)	8.9 (IQR: 1.1)
RASS 0/−1	1.3%	0.6%	7.4%	16.4%	7.5%
RASS −2/−3	29.3%	31.2%	32.6%	32.2%	31.6%
RASS −4/−5	68.8%	68.1%	58.8%	47.7%	59.3%
Kreatinin [mg/dL]	0.76 (IQR: 0.66)	0.83 (IQR: 1.09)	0.67 (IQR: 0.65)	0.4 (IQR: 0.3)	0.6 (IQR: 0.6)
IL-6 [pg/mL]	242 (IQR: 872)	187 (IQR: 621)	140 (IQR: 285)	73 (IQR: 113)	115 (IQR: 281)
PCT [ng/mL]	0.59 (IQR: 1.74)	0.48 (IQR: 1.92)	0.54 (IQR: 1.41)	0.51 (IQR: 1.11)	0.52 (IQR: 1.41)
Leukozyt [count/nL]	10.6 (IQR: 7.5)	10.5 (IQR: 6.8)	9.9 (IQR: 5.38)	8.55 (IQR: 4.7)	9.59 (IQR: 5.55)
LDH [U/L]	531 (IQR: 290)	528 (IQR: 295)	558 (IQR: 262)	487 (IQR: 219)	524 (IQR: 260)
patients with CRRT	11 [36.7%]	22 [56.4%]	18 [54.5%]	9 [40.9%]	60 [51.3%]
Oxygenator Exchange	3	19	48	40	110
Cannula change (*n* =)	2	5	3	4	14
minor bleedings (*n* =)	102	472	1194	652	2420
RBC transfusions (*n* =)	89	408	587	357	1441
prone positioning (*n* =)	120	150	187	61	518

Therapeutic and interventional characteristics structured according to the therapy duration of the included patients receiving VV-ECMO therapy. Data are presented as median (±interquartile range (IQR)) or as patient number [percentage] where applicable. Abbreviations: CRRT, continuous renal reverse therapy; Hb, hemoglobin; IL-6, interleukin-6; L, litre. LDH, lactate dehydrogenase; mg milligram; mL, millilitres; ng, nanogram; nL, nanolitres; PCT, procalcitonin; p_a_CO_2_, arterial carbon dioxide partial pressure; p_a_O_2_, arterial oxygen partial pressure, pg, picogram; RASS, Richmond Agitation-Sedation Scale; RBC, red blood concentrates; U, Units. * Gas flow through the oxygenator in liters per minute; # partial gas partial pressure measured before respectively shortly after initiation of VV-ECMO therapy.

## Data Availability

Raw data were generated at University Hospital of Frankfurt. Derived data supporting the findings of this study are available from the corresponding author A.N.F. on reasonable request.

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
