# Peer review of "Therapy and Outcome of Prolonged Veno-Venous ECMO Therapy of Critically Ill ARDS Patients"

_jcm, 2023, doi:10.3390/jcm12072499_

Round 1

Reviewer 1 Report

This manuscript reports the findings of a retrospective cohort study examining the relationship between duration of extracorporeal membrane oxygen (ECMO) support and survival. The key finding was that duration of ECMO therapy was not associated with mortality.

            This study addresses an important question faced by clinicians managing patients on ECMO, specifically if there is a time point at which therapy should be discontinued. The results may help provide support to clinicians making these decisions. However, the study faces specific challenges that limit the applicability of its results.

I would like to thank the authors for the opportunity to read their paper and hope that my comments will be of help.

Main concerns:

1.     The design of the study introduces immortal time bias (DOI: 10.1093/aje/kwm324), that is patients had to survive long-enough to experience a prolonged course of ECMO, consequently the study is biased to demonstrate that prolonged ECMO is not associated with increased mortality.

2.     Review of table 1 demonstrated significant difference between the exposure groups. Without adjustment for this underlying confounding, it is unclear if the lack of difference in mortality is related to healthier patients receiving longer ECMO durations or represents a true absence of difference associated with ECMO durations.

3.     The primary outcome is not defined. Does mortality mean ICU mortality, hospital mortality or mortality at a specific time point?

Specific suggestions/comments across each section:

Abstract:

1.     If space permits additional details related to the methods could be beneficial to the reader, such details could include:

a.     Definition of the four exposure durations (<14 days, 14-27, 28-49, 50+)

b.     Definition of survival (ICU, hospital, 90-day?)

c.     Statistical approach used to address confounding.

2.     Recommend the results include the raw data related to the primary outcome, i.e., proportion of deaths in each group, and the associated p-value demonstrating non-significance.

3.     Given the limitations inherent to observational data, consider softening the language of the conclusion, the statement “cannot be used” is definite and does not reflect the challenges of observational studies.

Materials and methods:

1.     It may be helpful to the audience to provide more details regarding the hospital at which the study was conducted as these details can allow the reader to determine if the clinical scenario is similar to the environment in which they work. Such details could include hospital size, ICU size, overall volume, ECMO volume, case-mix, etc.

2.     I see in the results that patients who received VA ECMO were excluded. Recommend that exclusion criteria be included in the methods section.

3.     Consider the addition of further subheadings in the methods section, for example:

a.     A section describing the therapies provided in ICU including ECMO therapy (rather than including a description of these therapies as part of the patient population)

b.     A section describing the exposure status.

c.     A section describing the outcome.

4.     It is unclear from the data collection section if data was collected in real time or if a data abstractor went back to the electronic recorded and extracted the relevant variables.

5.     Recommend defining the variables that were collected, i.e., what demographics, and the time points at which they were collected, i.e., is it daily ventilation parameters, every time a change is made, etc.

6.     It appears that no adjustments were made to address confounding or immortal time bias. Recommend expanding the statistical analysis to address the challenges inherent to observational studies.

Results:

1.     Table 1 combines baseline characteristics and clinical outcomes. The combination of these data points makes it challenging to read. Recommend splitting baseline characteristics and outcomes into different tables/figures.

2.     Recommend including duration of ventilation prior to ECMO initiation to table 1 given the impact of prolonged ventilation on survival. Also consider including ventilatory and oxygenation parameters that are discussed later in the results.

3.     The organization of the results section is challenging to follow. The primary outcome is present prior to a description of the study cohort. Likewise, baseline characteristics and outcomes and presented in the same section, i.e., age and mortality, then sex and mortality. Recommend revising the flow of the results section. Consider the first paragraph focused on a description of the cohort, then the second paragraph discussing the primary outcome (mortality).

4.     The results present a significant number of analyses of association (ECMO duration with mortality, age with mortality, sex with mortality, baseline O2 and CO2 with mortality, transfusions with mortality, transfusions with oxygenator change, etc.). The high number of analyses raises concern for identification of an association by chance. Recommend focusing on primary outcome and highlighting that all other analysis are hypothesis generating.

Discussion:

1.     Recommend the limitations section be expanded. Regardless of how confounding is addressed, I suspect unmeasured confounding will remain.

Minor comments:

Results:

1.     In table 1, the number within each group is presented as a row in the table. Consider moving the number into the header of each column rather than presenting as a data row.

Again, I would like to thank the authors for the opportunity to read their manuscript and hope that these comments are of help.

Reviewer 2 Report

Thank you for giving me tha opportunity to revise the manuscript. 

I would appreciate the following changes:

- extensive english editing

- Table reporting etiologies of ARDS

- Fine tuning of the conclusions looking at the role of identifying potential survivors after prolonged VV support
